# Demonstration of Spatial Modulation Using a Novel Active Transmitter Detection Scheme with Signal Space Diversity in Optical Wireless Communications

**DOI:** 10.3390/s22229014

**Published:** 2022-11-21

**Authors:** Tingting Song, Ampalavanapillai Nirmalathas, Christina Lim

**Affiliations:** Department of Electrical and Electronic Engineering, The University of Melbourne, Melbourne, VIC 3010, Australia

**Keywords:** optical wireless communications, spatial modulation, active transmitter detection, signal space diversity

## Abstract

Line-of-sight (LOS) indoor optical wireless communications (OWC) enable a high data rate transmission while potentially suffering from optical channel obstructions. Additional LOS links using diversity techniques can tackle the received signal performance degradation, where channel gains often differ in multiple LOS channels. In this paper, a novel active transmitter detection scheme in spatial modulation (SM) is proposed to be incorporated with signal space diversity (SSD) technique to enable an increased OWC system throughput with an improved bit-error-rate (BER). This transmitter detection scheme is composed of a signal pre-distortion technique at the transmitter and a power-based statistical detection method at the receiver, which can address the problem of power-based transmitter detection in SM using carrierless amplitude and phase modulation waveforms with numerous signal levels. Experimental results show that, with the proposed transmitter detection scheme, SSD can be effectively provided with ~0.61 dB signal-to-noise-ratio (SNR) improvement. Additionally, an improved data rate ~7.5 Gbit/s is expected due to effective transmitter detection in SM. The SSD performances at different constellation rotation angles and under different channel gain distributions are also investigated, respectively. The proposed scheme provides a practical solution to implement power-based SM and thus aids the SSD realization for improving system performance.

## 1. Introduction

Optical wireless communications (OWC) are a promising solution to cope with the challenges of the ever-increasing data volume and bandwidth brought by the emerging indoor wireless applications such as high-definition streaming in remote working, education, entertainment, etc. With easy installation via the Fiber to the Premises (FTTP) networks, the OWC system can support higher bandwidth wireless transmission with scalability than its radio frequency (RF) counterpart, regardless of RF interference and spectrum regulations [1,2,3]. Particularly, line-of-sight (LOS) OWC systems are widely studied nowadays due to their superior data rates and better power efficiency [4]. OWC is also identified as one of the potential enabling technologies for the upcoming sixth-generation (6G) wireless communications due to its ultra-high scalable bandwidth for achieving broadband wireless connectivity [5,6,7]. As partial obstruction of the line-of-sight (LOS) optical wireless beams typically occurs upon the deployment of the OWC system, maintaining a competitive high data rate OWC transmission is a significant challenge. 

Given that LOS optical channel is susceptible to being obstructed by the in-between small opaque objects or moving users, a resilient OWC system can be established with the redundant LOS links provided by additional transmitters/receivers [8]. For indoor OWC applications, additional transmitters are favorable to deploy as they have more flexibility for centralized control and also enable a compact, power-efficient end user without introducing the complex receiver configuration. As differences in channel gain across multiple LOS links are often observed, multi-transmitter deployment can be further leveraged to boost the system throughput in addition to providing diversity [9].

Spatial multiplexing (SMux) and spatial modulation (SM) are such data rate boost techniques that perform better under channel gain imbalance [10]. Compared to SMux, SM is reported to be more robust in combating the highly correlated channels that typically occur in the LOS indoor optical wireless links, with lower complexity RF chain configurations [11]. Even so, high channel correlation still degrades the accurate detection of the active transmitter in SM. To address this issue, various channel coding schemes have been proposed [12,13,14], where the “encoder-decoder” configuration increases the computation complexity of the system. On the other hand, precoding design applied to SM at the transmitter is another effective approach to enhance the robustness of active transmitter detection in highly correlated channels, with partial or full channel state information (CSI) pre-known [15,16].

However, SM itself does not achieve transmit diversity for enhancing link robustness [11]. The signal constellation dimension has been further explored to offer diversity [17,18] and simultaneously take advantage of the simplified RF channel activation in SM without introducing extra antenna matrix projection [19]. It is reported that a simple signal space diversity (SSD) technique using phase rotation can be integrated with SM [20] to enhance diversity through the complex Rayleigh fading channels. For the typical intensity modulation/direct detection (IM/DD)-based indoor OWC transmission, however, a different channel condition with real-valued channel gains is expected [21]. Therefore, this SSD technique has been further modified and comprehensively investigated in our previous work to show a distinct diversity performance [22,23].

Note that, all the above SSD investigations were based on the assumption of perfect detection of the active transmitter in SM. On the other hand, the performance of active transmitter detection can also be significantly affected by the high channel correlation in practical scenarios, which further affects the recovery of carrierless amplitude and phase (CAP) modulation signals involving many signal levels. To address the active transmitter detection issue in SM incorporating SSD, we have theoretically proposed a joint maximum-a-posteriori (MAP) estimation algorithm [24]. The joint MAP algorithm is implemented based on the conditional probabilities of two consecutive transmitted CAP sample-level distributions in a symbol for active transmitter detection. However, as the distributions of CAP sample levels can be complicated at different constellation rotation angles for offering SSD, this algorithm exhibits a relatively high computation complexity. In this paper, instead of using the conditional probabilities of transmitted CAP sample levels, we propose a novel active transmitter detection scheme in SM based on the observed CAP signal characteristics under various channel gain distributions. The transmitter detection scheme consists of the transmitted signal pre-distortion technique and a novel received power-based statistical detection method. Experimental results show that, using the proposed transmitter detection scheme, the previously proposed SSD technique can be effectively realized with an improved system throughput brought by SM.

Therefore, spatial modulation, as a low RF-chain complexity spatial reuse technique, can boost the data rate under such channel gain imbalance, where an effective detection of active transmitter enables SM implementation. Together with the signal space diversity technique, the performance of the OWC system can be further enhanced, which is exciting for achieving a reliable high-speed wireless transmission for 6G networks and beyond.

## 2. Principle of SM Using a Novel Active Transmitter Detection Scheme with SSD

### 2.1. System Architecture

The architecture of the overall indoor OWC system is shown in Figure 1, which includes the digital signal flows highlighted in the yellow box in Figure 1a and the multi-transmitter-single-receiver-based indoor OWC link configuration illustrated in Figure 1b. At the transmitter side, the proposed digital signals for implementing SM with SSD are generated offline and fed into the optical transmitters Tx*_i_* for optical modulation, where *i* ∈{1,2} is assumed here to simplify the analysis. These optical transmitters are centralized coordinated via the fiber network, where the generated spatial bits in SM control the optical transmitter activation. Thus, the optical signals from different transmitters, featured by distinct optical power, are projected into the indoor free space in turns at different SM time slots. The divergent free space optical (FSO) signals are then partially captured and optically detected by the single optical receiver Rx for further offline signal recovery.

### 2.2. Digital Implementation of SM with SSD

To implement SM with SSD, the original binary signal bits are divided into many subsets, followed by two consecutive subsets grouped exclusively for implementing SSD, as exemplified by the two underbraced subsets “100” and “010” in Figure 1a. The bits in each subset are also divided to be used for SM (called spatial bits for short, exemplified by blue numbers) and signal modulation (exemplified by purple numbers). Here, the SM approach, as shown in the gray inset, is employed with spatial bit = 0 and 1 for activating Tx_1_ and Tx_2_, respectively, where spatial bits differ in a signal group to enable the best SSD diversity performance [23]. As for the signal modulation, the two-subset-based signal group is used to provide SSD via a two-step signal constellation transformation. Thus, as a prerequisite, each pair of signal constellation points, such as “−1 − *j*” and “1 − *j*”, must be created via the quadrature amplitude modulation (QAM). The first step of constellation transformation is through the constellation rotation by multiplying “e*^jφ^*” in the constellation pair. As two spatial bits differ in a signal group, diversity interleaving with a 100% interleaving ratio is then performed as the second step by exchanging the imaginary part of one signal with the real part of the other signal. After the constellation transformation, CAP modulation is utilized to create real-valued signals for optical modulation. To enhance the robustness of spatial bit detection, here, the CAP signals are further pre-distorted without knowing much CSI. The pre-distortion technique will be described in detail in Section 2.3.

After optical signal detection, digital signal bits are obtained. The spatial bits that indicate the active transmitter are firstly recovered via a novel power-based statistical method, which will also be further introduced in Section 2.3. Since the CAP signal levels are significantly affected by the distinct channel gains from different transmitters, such differences in channel gain can be compensated separately to recover the whole CAP signals in terms of each channel, i.e., Tx_1_’s channel gain is compensated to that of Tx_2_ to recover the whole CAP signals in terms of signal levels from Tx_2_, and vice versa. After two sets of the whole CAP signals are recovered in terms of signal levels from Tx_1_ and Tx_2_, respectively, the signals from different transmitters can be picked up, respectively, and combined to perform the inverse transformation of the signal constellation, including diversity deinterleaving, K-means decision and QAM demapping. Here, K-means clustering is used for the signal decision to combat the nonlinearity introduced by pre-distortion and channel-gain-compensated CAP signal recovery. The recovered spatial bits and signal modulation bits are then combined to finalize the signal recovery.

### 2.3. The Novel Active Transmitter Detection Scheme

The proposed novel active transmitter detection scheme consists of the signal pre-distortion technique at the transmitter and a novel power-based statistical method for active transmitter detection at the receiver. The signal pre-distortion technique shapes the CAP signal power at the transmitter to facilitate active transmitter detection. The power-based statistical method is then implemented to reduce the transmitter detection error statistically.

#### 2.3.1. Signal Pre-Distortion Technique

The signal pre-distortion technique is applied after CAP modulation at the transmitter side, which can be described in Equations (1) and (2) as:(1)$$\begin{array}{c} \{\begin{array}{c} S_{1}=S_{1}\times \rho \\ S_{2}=\frac{S_{2}}{\rho }\times \tau \end{array}\;\;\;\mathrm{when}\;\mathrm{both}\;\left(\max\left(\left|S_{1}\right|\right)<\max\left(\left|S_{2}\right|\right)\right)\;\mathrm{and}\;\left(h_{1}>h_{2}\right)\;\mathrm{are}\;\mathrm{met},\;\mathrm{with}\;\rho =\sqrt{\frac{\max\left(\left|S_{2}\right|\right)}{\max\left(\left|S_{1}\right|\right)}} \end{array}$$
(2)$$\begin{array}{c} \{\begin{array}{c} S_{1}=\frac{S_{1}}{\rho }\times \tau \\ S_{2}=S_{2}\times \rho \end{array}\;\;\;\;\mathrm{when}\;\mathrm{both}\;\left(\max\left(\left|S_{1}\right|\right)>\max\left(\left|S_{2}\right|\right)\right)\;\mathrm{and}\;\left(h_{1}<h_{2}\right)\;\mathrm{are}\;\mathrm{met},\;\mathrm{with}\;\rho =\sqrt{\frac{\max\left(\left|S_{1}\right|\right)}{\max\left(\left|S_{2}\right|\right)}} \end{array}$$
where $S_{1}$ and $S_{2}$ are two consecutive CAP signal symbols that form an exclusive pair, with several samples included in each CAP symbol due to CAP pulse shaping. Additionally, *τ* is the suppress ratio with τ∈(0, 1), and $h_{1}$ and $h_{2}$ are the total channel gains of two spatial channels, i.e., the channel gains from optical modulation to the optical detection, which pass through the electrical–optical–electrical conversion. Here, it is noted that except for the pre-knowledge of the allocation of the maximum channel gain across the spatial channels, there is no other explicit CSI that needs to be pre-known, which simplifies the implementation of the pre-distortion technique.

Figure 2a shows the signal flow for implementing pre-distortion. The pre-knowledge of the maximum channel gain allocation is obtained through initial channel training and then fed back to the transmitter side to enable pre-distortion. If $h_{1}>h_{2}$ and $\max\left(\left|S_{1}\right|\right)<\max\left(\left|S_{2}\right|\right)$ are both met, pre-distortion is conducted using Equation (1); if $h_{1}<h_{2}$ and $\max\left(\left|S_{1}\right|\right)>\max\left(\left|S_{2}\right|\right)$ are both met, pre-distortion is conducted using Equation (2). Otherwise, the CAP symbol pair remains intact. Figure 2b shows an example of the generation of a pre-distorted CAP waveform. The blue, yellow, red and green dashed boxes in the CAP waveform indicate the range of implementing signal pre-distortion, where each dashed box contains one CAP symbol pair (oversampling factor = 4 for each symbol).

The pre-distortion technique is proposed based on the observation of CAP signal characteristics at the receiver. Due to the many signal levels possessed by CAP, the high signal levels in CAP with lower channel gain may be even larger than the low signal levels in CAP with higher channel gain, leading to a false active transmitter detection. The pre-distortion technique tackles this issue by limiting the high signal levels in CAP with lower channel gain while enhancing the low signal levels in CAP with higher channel gain. Although some nonlinearity is introduced in this technique, it still significantly improves the performance of resolving the active transmitter at the receiver.

#### 2.3.2. Power-Based Statistical Method for Active Transmitter Detection

The power-based statistical method for active transmitter detection is expressed in Equations (3) and (4):(3)$$\begin{array}{c} Z\left(t,j\right)=\langle a,\;\mathrm{sort}\left(\left|x_{t}^{j}\right|\right)\rangle +b\times \left(\max\left(x_{t}^{j}\right)-\min\left(x_{t}^{j}\right)\right)+c\times \sigma \left(x_{t}^{j}\right) \end{array}$$
where Z(t,j) is the power-based statistical function of the *j-*th symbol at the two-symbol-based time slot *t*, where *j*∈{1,2} since two symbols form a signal group to provide SSD. Besides, $x_{t}^{j}$ is the received signal sample vector of the *j-*th symbol at the two-symbol-based time slot *t* with each vector length = oversampling factor, $\mathrm{sort}\left(\left|x_{t}^{j}\right|\right)$ is the absolute value of $x_{t}^{j}$ arranged in descending order, a is the weight coefficient vector obtained via training with the reference of ideal active transmitter detection, 〈·〉 is the inner product, b and c are the weight coefficients also obtained via training, and $\sigma \left(x_{t}^{j}\right)$ is the standard deviation of $x_{t}^{j}$.

When Z(t,1) and Z(t,2) are obtained, the spatial bits I(t) in a signal group can be recovered as:
(4)$$I\left(t\right)=\left\{\begin{array}{cc} \left(0,1\right) & \;\mathrm{when}\;\mathrm{either}\;\left(Z\left(t,1\right)>Z\left(t,2\right)\;\mathrm{and}\;h_{1}>h_{2}\right)\;\mathrm{or}\;\left(Z\left(t,1\right)<Z\left(t,2\right)\;\mathrm{and}\;h_{1}<h_{2}\right)\;\mathrm{are}\;\mathrm{met}\;\\ \left(1,0\right) & \;\mathrm{Otherwise}.\; \end{array}\right.$$

The proposed power-based statistical method is based on the observation of received CAP signal characteristics and the channel gain conditions. In Equation (3), received signal power distribution is expressed statistically in three terms. The first inner product term gives a weighted symbol level since the contribution of each sample differs to represent a symbol. The second term indicates the range of sample levels in each symbol. The third term describes the amount of variation in the samples in each symbol. This received signal power distribution is then used to recover the spatial bits in Equation (4) with reference to the measured total channel gains. An example is given to illustrated how it works: when the values of Z(t,1) and Z(t,2) coincide with the reference total channel gains, i.e.,$\;Z\left(t,1\right)>Z\left(t,2\right)\;\mathrm{and}\;h_{1}>h_{2}$, it indicates the first spatial bit is from Tx_1_ and the second is from Tx_2_. Thus, the pair of spatial bits (0,1) is recovered. Otherwise, when the values of Z(t,1) and Z(t,2) is opposite to the reference total channel gains, the pair of spatial bits (1,0) is recovered.

This power-based method with three statistical terms enables a comprehensive description of the signal power distribution and thus enhances the robustness of power-based active transmitter detection under noisy channel conditions.

## 3. Experimental Setup and Results

### 3.1. Experimental Setup

The proof-of-concept experiment of the novel active transmitter detection scheme in SM that provides SSD was conducted using the two-transmitter-single-receiver setup in Figure 3a, where the SM implementation is illustrated in the inset. The digital signals were generated offline using the proposed approach in Figure 1a. Then, instead of directly routing the pre-distorted CAP signals to each transmitter, zeros were inserted into these CAP signals at the time slots when that transmitter was inactive, representing null signal transmission. The length of zeros at each SM time slot is equal to the CAP oversampling factor. After that, these new packed digital signals were fed into an arbitrary waveform generator (AWG), with two channels of 10 GS/s AWG outputs routed to modulate two 12 GHz directly modulated lasers (DMLs) using wavelengths of 1549.6 nm and 1550.3 nm, respectively. After each DML, an optical attenuator was placed to differentiate optical power between two transmitters, thus, enabling SM transmission. Additionally, an optical delay line was utilized in one of the channels for the fine-tuning of channel synchronization (coarse-tuning is achieved by digitally delaying 180 samples in the “fast” channel). Through such optical fiber links, the optical signals were then emitted into the FSO channels respectively, with a 16° beam divergence and a maximum power of 3.5 dBm for each signal, which is below the laser safety regulation [4]. After a 0.5 m FSO transmission, partial optical signal power from two transmitters was collected and coupled into the photodetector using a lens system. The digital photodetector (PD) output was finally input into an oscilloscope for offline signal processing. A photo of the experimental setup is shown in Figure 3b. The physical parameters of the experimental setup are summarized in Table 1.

### 3.2. Results and Discussions

In the demonstration, ~7.5 Gbit/s data rate can be achieved using 2.5 GBaud/s 4-CAP modulation incorporating SM. The capability of the proposed active transmitter detection scheme in SM to provide SSD is shown in Figure 4a, where the channel gain difference ratio is $\frac{h_{2}}{h_{1}}=0.5$. Here, the ideal active transmitter detection is defined as the perfect detection of the spatial bits. The measured ideal system bit-error-rate (BER) is thus described as:(5)$$\begin{array}{c} B{\mathrm{ER}}_{\mathrm{ideal}}=\frac{\mathrm{Measured}\;\mathrm{bit}\;\mathrm{errors}\;\mathrm{of}\;\mathrm{CAP}\;\mathrm{signals}}{\mathrm{The}\;\mathrm{length}\;\mathrm{of}\;\mathrm{original}\;\mathrm{bits}}\; \end{array}$$
where the original bits contain CAP signal bits and spatial bits. On the other hand, in our proposed transmitter detection scheme, the measured system BER is described as:(6)$$\begin{array}{c} B{\mathrm{ER}}_{\mathrm{proposed}}=\frac{\mathrm{Measured}\;\mathrm{bit}\;\mathrm{errors}\;\mathrm{of}\;\left(\mathrm{CAP}\;\mathrm{signals}\;+\;\mathrm{spatial}\;\mathrm{bits}\right)}{\mathrm{The}\;\mathrm{length}\;\mathrm{of}\;\mathrm{original}\;\mathrm{bits}}. \end{array}$$

Thus, compared with the ideal and the proposed transmitter detection scheme at the best-performed constellation rotation, i.e., 45° rotation [23], shown as the red and black curves, respectively, our proposed transmitter detection scheme only results in a slightly worse measured BER, implying a low-level detection error of spatial bits. Moreover, as 0° constellation rotation provides no SSD, compared with blue and red curves, our proposed transmitter detection scheme outperforms as SSD is provided, which in turn indicates the detection error of spatial bits is less than the BER improvement brought by SSD. It can be seen that about 0.61 dB signal-to-noise-ratio (SNR) improvement is achieved. To further explore the SSD performance at different constellation rotations using our proposed scheme, the 30° constellation rotation performance is also studied as in the green curve. The result further verifies the previous theoretical result of SSD performance at different rotation angles where ideal SM performance is assumed [23], i.e., the SSD is enhanced with an increasing constellation rotation up to 45° under a specific channel gain difference ratio such as $\frac{h_{2}}{h_{1}}=0.5$, thus showing the robustness of our proposed transmission detection scheme to offer SSD.

The proposed active transmitter detection scheme in SM is also experimentally investigated under different channel gain distributions at the 45° constellation rotation, as shown in Figure 4b. An improved BER performance is observed with an increasing channel gain difference ratio from $\frac{h_{2}}{h_{1}}=0.4$ to $\frac{h_{2}}{h_{1}}=0.5$, which also follows the BER trend investigated in [23], i.e., a larger $\frac{h_{2}}{h_{1}}$ enhances the SSD performance. However, a different BER result is observed when the channel gain difference ratio is further increased to $\frac{h_{2}}{h_{1}}=0.66$. That is because, when $\frac{h_{2}}{h_{1}}=0.4$, a large difference in received signal power between two channels is expected, which facilitates the spatial bits recovery. In fact, here, the error ratio of the spatial bits and CAP signal bits (short for error ratio) is in the range of [1:40 1:15], showing the dominant role of CAP signal errors. Thus, the BER performance is similar to the ideal SSD performance (without considering SM error) [23]. When $\frac{h_{2}}{h_{1}}$ is increased to 0.66, a more negligible difference of two received signal power is observed, leading to a more demanding spatial bits recovery with the error ratio increase to the range of [1:2 1:1.2]. These increasing errors of spatial bits result in the overall performance of a slightly degraded BER in the scenario of $\frac{h_{2}}{h_{1}}=0.66$. In other words, with the SM implementation using our active transmitter detection scheme, a performance balance between SM and SSD is expected when the channel gain difference is smaller.

The suppress ratio *τ* in the pre-distortion technique is studied to illustrate the impact of pre-distortion in the proposed transmitter detection scheme, where τ=1 is taken as a reference to represent a limited pre-distortion scenario, as inferred from Equations (1) and (2). As shown in Figure 4c, an improved BER performance can be obtained with a decreasing *τ* from 1 to 0.5, which verifies the effectiveness of the pre-distortion technique. However, the BER improvement is much smaller when *τ* is decreased from 0.7 to 0.5. That is because more signal nonlinearity is introduced with a smaller *τ*. In fact, a smaller *τ* also reduces the signal power efficiency. Therefore, a tradeoff of *τ* value must be made for optimized system performance. Here, τ= 0.5 is applied in our experiment.

Note that, here, our proposed SSD is provided using CAP modulation. In fact, it has also been theoretically reported that SSD can be provided using OFDM modulation [25]. However, when SM is incorporated with the power difference between channels, we found that under the transmitted optical power constraint, the received optical power budget for CAP and OFDM is lower in the experiment. Compared with CAP modulation, OFDM has lower energy efficiency and higher PAPR [26], which is more demanding for implementation under a tight received optical power budget. Moreover, a customized pre-distortion technique for differentiating active transmitters based on OFDM signal characteristics needs to be further considered with equalization adopted [27].

## 4. Conclusions

A novel active transmitter detection scheme in spatial modulation that consists of a transmitted signal pre-distortion technique and a received power-based statistical detection method is proposed to be incorporated with a signal space diversity technique. Experimental results have shown that the proposed transmitter detection scheme can effectively recover the power-based spatial bits from the numerous-level-based CAP waveforms, thus realizing an improved system data rate brought by SM which is ~7.5 Gbit/s. Meanwhile, SSD can be offered with about 0.61 dB SNR improvement. In addition, improved BER performance with an increasing constellation rotation angle is verified when the channel gain difference ratio equals to 0.5. Furthermore, it is indicated that by using our transmitter detection scheme, a tradeoff between SM and SSD performances is expected with a decreasing channel gain difference, which is different from the ideal SSD performance without considering SM. As for the effectiveness of the pre-distortion technique, the BER improvement can be significant by selecting an appropriate suppress ratio without introducing much nonlinearity. The proposed active transmitter detection scheme provides a practical solution to implement power-based SM that incorporates SSD for pursuing a better system BER with enhanced system throughput.

## Figures and Tables

**Figure 1 sensors-22-09014-f001:**
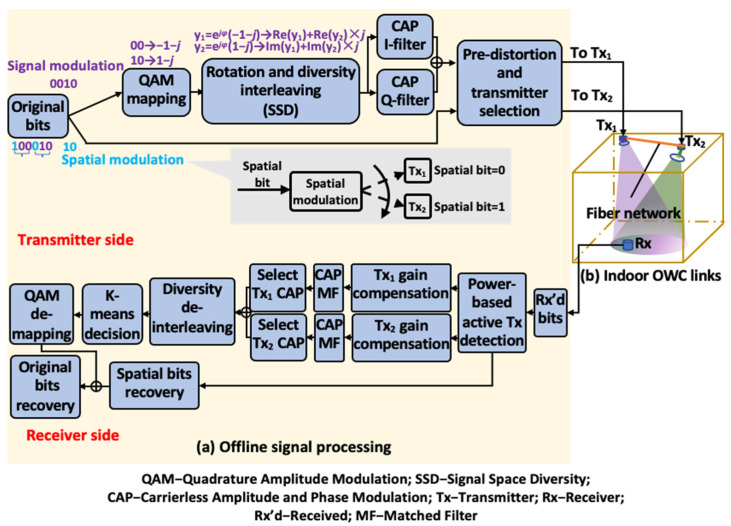
The architecture of the overall indoor OWC system.

**Figure 2 sensors-22-09014-f002:**
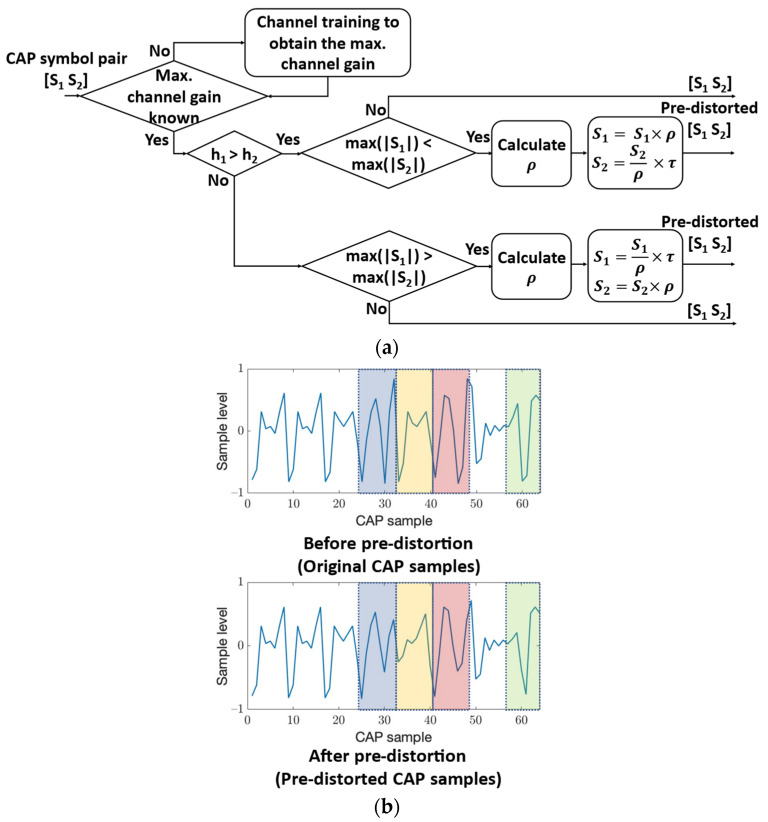
Signal pre-distortion technique. (**a**) Signal flow for pre-distortion. (**b**) Example of the generation of a pre-distorted CAP waveform.

**Figure 3 sensors-22-09014-f003:**
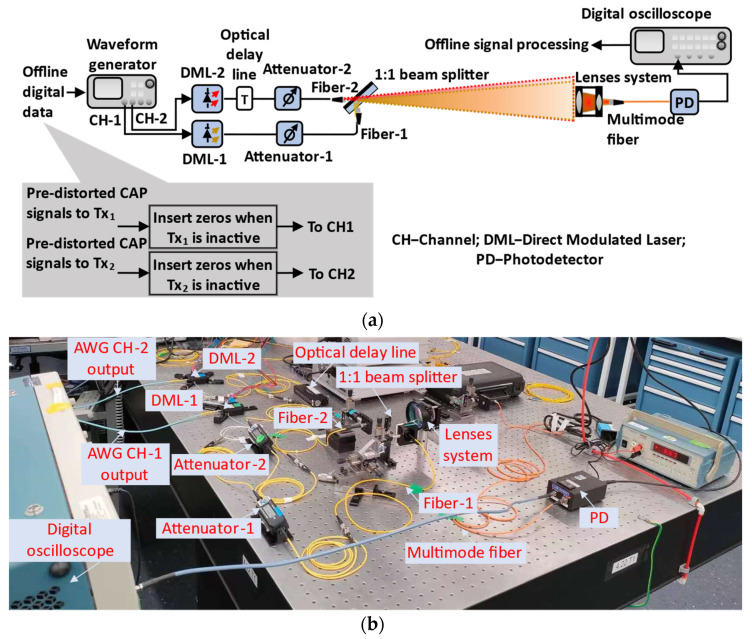
Experimental setup. (**a**) Schematic diagram. (**b**) Photo.

**Figure 4 sensors-22-09014-f004:**
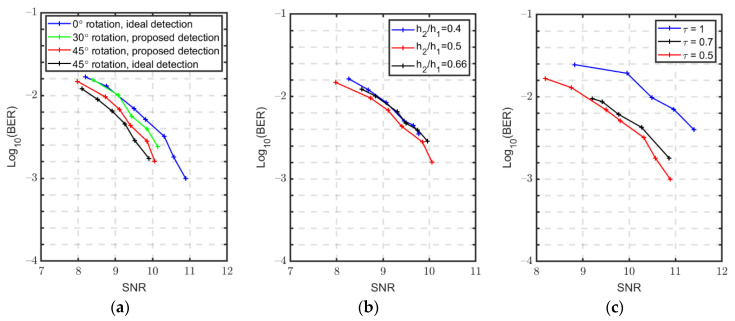
BER of the SM with SSD using the proposed novel active transmitter detection scheme. (**a**) ideal detection vs. the proposed detection. (**b**) Different channel gain distributions. (**c**) Different suppress ratios *τ*.

**Table 1 sensors-22-09014-t001:** Physical parameters of the experimental setup.

Device Name	Model	Parameters
Waveform generator	Tektronix AWG 7102	10 GS/s 4-CAP (oversampling factor = 4) Vpp = 1 V Sample length = 20,000
Directly modulated laser (DML)	Gooch and Housego EM657	1549.6 nm and 1550.3 nm
Optical delay line	General Photonics Corp. VariDelayTM	0.2 cm (13.3 ps)
Attenuator	EigenLight power monitor 420 WDM	Maximum reading = 3.5 dBm
Photodetector	Discovery Semiconductor Inc. DSC-R402	DC Responsivity @ 1550 nm: 0.84 A/W −3 dB Bandwidth: 10 GHz
Oscilloscope	Tektronix TDS6154C	40 GS/s Oversampling factor = 4 Record length = 200,000

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
