# Peer review of "Demonstration of Spatial Modulation Using a Novel Active Transmitter Detection Scheme with Signal Space Diversity in Optical Wireless Communications"

_sensors, 2022, doi:10.3390/s22229014_

Round 1
Reviewer 1 Report
1. In the Introduction, the authors could better differentiate this work with respect to their previous; also, they could limit the broader descriptions and dedicate more space to the impact of this specific work.
2. Section 2.3, which is the core of this work, could be better explained by means of simulation examples or including signal flow graphs for the processing step. This is typically useful for DPD schemes. For example, it is not clear how the feedback is implemented and how the predistorted waveforms are synthesized.
3. Is there any mutual couplings between the ways or particular linear equalization procedures that should be taken into account? This is a classical problem in RF arrays as shown in recent publications and conferences, e.g. Mengozzi et al. "Over-the-Air Digital Predistortion of 5G FR2 Beamformer Array by exploiting Linear Response Compensation" IMS2 022. Please comment.
4. Could the authors list the element and manufacturer of the instruments and devices used in the setup? That is useful for reproducing the results. Adding a photo of the setup would be beneficial.
5. The authors should improve Section 3.2. Also, please note that Section 4 does not exist. The overall results seem to be quite a minor improvement, much less than a dB in BER. Also there's some lack of comparison with other methods and no testing for different modulation schemes. Please provide info on the modulation scheme, bandwidth, etc.
Reviewer 2 Report
The authors have investigated an active transmitter detection scheme, incorporated with the signal space diversity technique, to improve the performance of the optical wireless communication system. This system includes a signal pre-distortion technique at the transmitter side and a power-based statistical detection at the receiver side. The manuscript is interesting and useful for the application and the development of optical wireless communications. The paper is acceptable to be published in Sensors, provided the following issue can be addressed
- Some abbreviations should be clarified when they appear for the first time.
- Add more details in the table to describe the physical parameters of the considered testbed.
- Improve the quality of Figure 3, since this is the main result of the paper but now it is not clear enough.
- Add some discuss on the impact of different modulation formats on the performance of the investigated scheme.
- Add some discussion either in the Discussion or the Introduction Section regarding other potential distortions that need to be considered in more practical secure communication networks.
See e.g.
C Jin et al., Nonlinear coherent optical systems in the presence of equalization enhanced phase noise, Journal of Lightwave Technology, 2021.
F Hu et al., High-speed visible light communication systems based on Si-substrate LEDs with multiple superlattice interlayers, PhotoniX, 2021.
M Baier et al., Integrated transmitter devices on InP exploiting electro-absorption modulation, PhotoniX, 2020.
Round 2
Reviewer 1 Report
I thank the authors for their reply.
